# Formation of the first three gravitational-wave observations through isolated binary evolution

Simon Stevenson[1], Alejandro Vigna-Gómez[1], Ilya Mandel[1], Jim W. Barrett[1], Coenraad J. Neijssel[1], David Perkins[1] & Selma E. de Mink[2]

During its first four months of taking data, Advanced LIGO has detected gravitational waves from two binary black hole mergers, GW150914 and GW151226, along with the statistically less significant binary black hole merger candidate LVT151012. Here we use the rapid binary population synthesis code COMPAS to show that all three events can be explained by a single evolutionary channel—classical isolated binary evolution via mass transfer including a common envelope phase. We show all three events could have formed in low-metallicity environments ($Z = 0.001$) from progenitor binaries with typical total masses $\gtrsim 160 M_\odot$, $\gtrsim 60 M_\odot$ and $\gtrsim 90 M_\odot$, for GW150914, GW151226 and LVT151012, respectively.

[1] School of Physics and Astronomy, University of Birmingham, Edgbaston, Birmingham, West Midlands B15 2TT, UK. [2] Anton Pannekoek Institute for Astronomy, University of Amsterdam, 1090 GE Amsterdam, The Netherlands. Correspondence and requests for materials should be addressed to S.S. (email: simon.stevenson@ligo.org).

The Advanced Laser Interferometer Gravitational-wave observatory (aLIGO)[1] has confidently observed gravitational waves (GWs) from two binary black hole (BBH) mergers, GW150914 (ref. 2) and GW151226 (ref. 3). The BBHs merger candidate LVT151012 is less statistically significant, but has a >86% probability of being astrophysical in origin[4,5].

GW150914 was a heavy BBHs merger, with a well-measured total mass $M = m_1 + m_2 = 65.3 \pm^{4.1}_{3.4} \, M_\odot$ (refs 5,6), where $m_{1,2}$ are the component masses. Several formation scenarios could produce such heavy BBHs. These include the following: the classical isolated binary evolution channel we discuss in this study[7–9], including formation from population III stars[10]; formation through chemically homogeneous evolution in very close tidally locked binaries[11–13]; dynamical formation in globular clusters[14–16], young stellar clusters[17] or galactic nuclei[18,19]; or even mergers in a population of primordial binaries[20,21]. One common feature of all GW150914 formation channels with stellar-origin black holes (BHs) is the requirement that the stars are formed in sub-solar metallicity environments, to avoid rapid wind-driven mass loss, which would bring the remnant masses below $30 M_\odot$ (refs 22,23); see Results and Abbott et al.[5,24] for further discussion.

We are developing a platform for the statistical analysis of observations of massive binary evolution, Compact Object Mergers: Population Astrophysics and Statistics (COMPAS). COMPAS is designed to address the key problem of GW astrophysics: how to go from a population of observed sources to understanding uncertainties about binary evolution. In addition to a rapid population synthesis code developed with model-assumption flexibility in mind, COMPAS also includes tools to interpolate model predictions under different astrophysical model assumptions, astrostatistics tools for population reconstruction and inference in the presence of selection effects and measurement certainty, and clustering tools for model-independent exploration.

Here we attempt to answer the following question: can all three LIGO-observed BBHs have formed through a single evolutionary channel? We use the binary population synthesis element of COMPAS to explore the formation of the observed systems through the classical isolated binary evolution channel[25] via a common envelope (CE) phase[26]. We show that GW151226 and LVT151012 could have formed through this channel in an environment at $Z = 10\% Z_\odot$ (with $Z_\odot \equiv 0.02$) from massive progenitor binaries with a total zero-age main-sequence (ZAMS) mass $\gtrsim 65 \, M_\odot$ and $\gtrsim 95 \, M_\odot$, respectively.

These BBHs could also originate from lower-mass progenitors with total masses $\gtrsim 60 \, M_\odot$ and $\gtrsim 90 \, M_\odot$, respectively, at metallicity $Z = 5\% Z_\odot$, where the same channel could have formed GW150914 from binaries with a total ZAMS mass $\gtrsim 160 \, M_\odot$. At low metallicity, this channel can produce merging BBHs with significantly unequal mass ratios: more than 50% of BBHs have a mass ratio more extreme than 2 to 1 at $Z = 10\% Z_\odot$.

## Results

**Forming GW151226 and LVT151012.** For relatively low-mass GW events, the GW signal in the aLIGO-sensitive frequency band is inspiral-dominated and the chirp mass $\mathcal{M} = M q^{3/5} (1+q)^{-6/5}$ is the most accurately measured mass parameter, while the mass ratio $q = m_2/m_1$ cannot be measured as accurately (see Fig. 4 of Abbott et al.[5]). The 90% credible intervals on these for GW151226 and LVT151012 are $8.6 \leq \mathcal{M}/M_\odot \leq 9.2$, $q \geq 0.28$ and $14.0 \leq \mathcal{M}/M_\odot \leq 16.5$, $q \geq 0.24$, respectively[5]. For more massive events, the ringdown phase of the GW waveform makes a significant contribution and the most accurately measured mass parameter is the total mass $M$. For GW150914, $M = 65.3 \pm^{4.1}_{3.4} \, M_\odot$ (refs 5,6), with mass ratio $q \geq 0.65$.

We simulate events at 10% solar ($Z = 0.002$) and 5% solar ($Z = 0.001$) metallicity using the Fiducial model assumptions (see Methods). We select binaries which fall within the 90% credible interval on total (chirp) BBH mass and with $q$ above the 90% credible interval lower bound for GW150914 (GW151226 and LVT151012). In all cases, we select only BBHs that merge within the Hubble time. Systems satisfying these conditions are shown in Fig. 1. The upper panel shows BBHs formed at 10% solar metallicity, whereas the lower panel shows those formed at 5% solar metallicity. The BH mass of the initially more massive star is labeled as $M_1^{BH}$ and that of the initially less massive star as $M_2^{BH}$.

In the left hand column of Fig. 1, we show the ZAMS masses of possible progenitors of these events. Progenitors of the events are separated in ZAMS masses apart from rare systems that start on very wide orbits, avoiding mass transfer altogether, but are brought to merger by fortuitous supernova kicks. These systems do not lose mass through non-conservative mass transfer and can therefore form more massive binaries from lower mass progenitors—the LVT151012 outlier progenitor in the lower left corner of the bottom left panel of Fig. 1 was formed this way.

Massive stars have high mass loss rates; for example, at solar metallicity, massive stars could lose tens of solar masses through winds even before interacting with their companion. We find, in agreement with Abbott et al.[24] and Belczynski et al.[7], that it is not possible to form GW150914 or LVT151012 through classical isolated binary evolution at solar metallicity. GW151226 lies at the high-mass boundary of BBHs that can be formed at solar metallicity.

GW151226 is consistent with being formed through classical isolated binary evolution at 10% solar metallicity from a binary with total mass $65 \lesssim M/M_\odot \lesssim 100$ (see upper left panel of Fig. 1). LVT151012 is also consistent with being formed at 10% solar metallicity from binaries with initial total masses $95 \lesssim M/M_\odot \lesssim 125$. Typical progenitors have a mass ratio close to unity (median $q = 0.75$), with an initial orbital period of $\sim 500$ days.

GW150914 could have formed through isolated binary evolution at metallicities $Z \lesssim 5\% Z_\odot$ from binaries with initial total mass $\gtrsim 160 \, M_\odot$ (see lower left panel of Fig. 1). Although this mass range is similar to that found by others who investigated the formation of GW150914 through isolated binary evolution at low metallicities[7–9], we note that, unlike Eldridge and Stanway[8], we do not require fortuitous supernova kicks resulting in high eccentricity to form this binary at $Z = 5\% Z_\odot$. We identify the same main evolutionary channel (see Fig. 2) as Belczynski et al.[7]. We find that GW151226 and LVT151012 are also consistent with forming through this channel at lower metallicity, from initially lower mass binaries. For example, the total progenitor binary mass range for forming GW151226 reduces from $65 \lesssim M/M_\odot \lesssim 100$ at 10% solar metallicity to $60 \lesssim M/M_\odot \lesssim 90$ at 5% solar metallicity, demonstrating a degeneracy in the ZAMS masses and metallicity inferred in our model due to the dependence of mass loss rates on metallicity.

We find that the chirp masses of GW151226 and LVT151012 lie near the peak of the mass distribution of BBHs mergers formed at 10% solar metallicity which are observable by aLIGO. There remains significant support for both systems at 5% solar metallicity. GW150914 cannot be formed at 10% solar metallicity in our model and remains in the tail of the total mass distribution at 5% solar, which is the highest metallicity at which we form significant numbers of all three event types in the Fiducial model. Events like GW150914 are much more common at 1% solar metallicity.

At $Z = 5\% Z_\odot$, the more massive BH is formed from the initially more massive star in $\sim 90\%$ of systems.

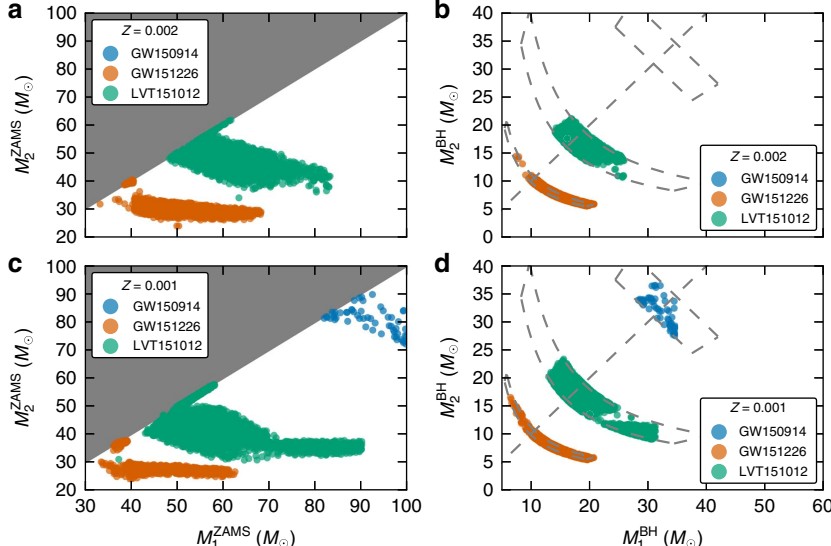

**Figure 1 | Masses of BBHs observed by aLIGO and their progenitors.** Each point in the plots represents one system in our simulations. (**a**) Zero-age main sequence (ZAMS) masses $M_1^{ZAMS}$ and $M_2^{ZAMS}$ for GW150914 (blue—no events), GW151226 (orange) and LVT151012 (green) progenitors at $Z = 10\%Z_\odot = 0.002$. We define $M_1^{ZAMS} > M_2^{ZAMS}$ and so shade the non-allowed region grey. (**b**) Final BH masses $M_1^{BH}$ and $M_2^{BH}$ for merging BBHs consistent with GW150914, GW151226 and LVT151012 formed at $Z = 10\%Z_\odot$. The grey diagonal dashed line shows $M_1^{BH} = M_2^{BH}$. The constraints we use to determine whether a merging BBHs is similar to one of the observed GW events are shown in grey and described in Results. (**c**) ZAMS masses $M_1^{ZAMS}$ and $M_2^{ZAMS}$ for GW150914, GW151226 and LVT151012 progenitors at the lower metallicity $Z = 5\%Z_\odot = 0.001$. The progenitor masses required to produce GW151226 and LVT151012 decrease, and we are able to produce GW150914. (**d**) Final BH masses $M_1^{BH}$ and $M_2^{BH}$ for GW150914, GW151226 and LVT151012 BBHs formed from 5% solar metallicity progenitors. The panels of this figure are formatted to be comparable to Fig. 4 in Abbott et al.[5].

Interestingly, low metallicities can produce significantly unequal mass ratios. For example, the median mass ratio of merging BBHs is $\sim 0.5$ at 10% solar metallicity. The high fraction of merging BBHs with low mass ratios at low metallicities is a general trend; this agrees with Fig. 9 of Dominik et al.[27], who do not, however, discuss this effect. A GW detection of a heavy BBHs with an accurately measured low mass ratio could indicate formation in a lower metallicity environment and not necessarily dynamical formation as suggested in Abbott et al.[5]

The significant fraction of low mass-ratio mergers at low metallicity arises due to a combination of effects. The maximum BH mass for single stars is a function of metallicity (for example, Fig. 6 of Spera et al.[23]), with more massive BH formed at lower metallicities due to reduced mass loss. Therefore, for a given observed chirp mass, more unequal BH can be formed at low metallicity. A second effect comes from the difference in the onset of the first episode of mass transfer, which is key for determining the mass of the remnant. The dependence of stellar radius on metallicity[28] means that stars with lower metallicity experience their first episode of mass transfer in a more evolved phase of their evolution for a given initial orbital separation[29]. They thus lose less mass when the hydrogen envelope is stripped, again allowing for more unequal remnants.

**Typical evolutionary pathway of GW151226.** In Fig. 2, we show the evolution in time of the masses, stellar types and orbital period of typical progenitors of all three observed GW events. Progenitors of all three systems follow the same typical channel. Here we describe the evolution of a typical 10% solar metallicity progenitor of GW151226 (solid orange line in Fig. 2); it is shown graphically in Fig. 3.

The binary initially has two high-mass main-sequence O stars, a primary of $\sim 64M_\odot$ and a $\sim 28M_\odot$ companion with an initial orbital period of $\sim 300$ days. The primary expands at the end of

its main sequence evolution, fills its Roche lobe and initiates mass transfer as a $\sim 60M_\odot$ Hertzsprung-Gap (HG) or core helium-burning (CHeB) star (case B or C mass transfer), donating its $\sim 36M_\odot$ hydrogen-rich envelope to the secondary, which accretes only $\sim 3M_\odot$ of it. This leaves the primary as a stripped naked helium star (HeMS) of $\sim 25M_\odot$. After evolving and losing a few solar masses through stellar winds, the primary collapses to a BH of $\sim 19M_\odot$ through almost complete fallback.

The secondary continues evolving and initiates mass transfer as a CHeB star of $\sim 30M_\odot$. This mass transfer is dynamically unstable and leads to the formation and subsequent ejection of a CE. The CE ejection draws energy from the orbit and results in significant orbital hardening: the orbital period is reduced by $\sim 3$ orders of magnitude as can be seen in the lower right panel of Fig. 2. The secondary, which becomes a HeMS star of $\sim 11 M_\odot$ after the ejection of the envelope, eventually collapses to a $\sim 6 M_\odot$ BH; the supernova kick drives the binary to an eccentricity of $\sim 0.5$. Finally, the binary merges through GW emission in $\sim 100$ Myrs.

A few per cent of our BBHs progenitors form through a variant of this channel involving a double CE. This variant involves two nearly equal mass ZAMS stars, which first interact during the CHeB phase of their evolution, initiating a double CE that brings the cores close together. This is followed by both stars collapsing into BH and merging through GW emission.

## Discussion

We have explored whether all of the GW events observed to date could have been formed through classical isolated binary evolution via a CE phase. All three observed systems can be explained through this channel under our Fiducial model assumptions. Forming all observed GW events through a single formation channel avoids the need to fine tune the merger rates from the very different evolutionary channels discussed in the

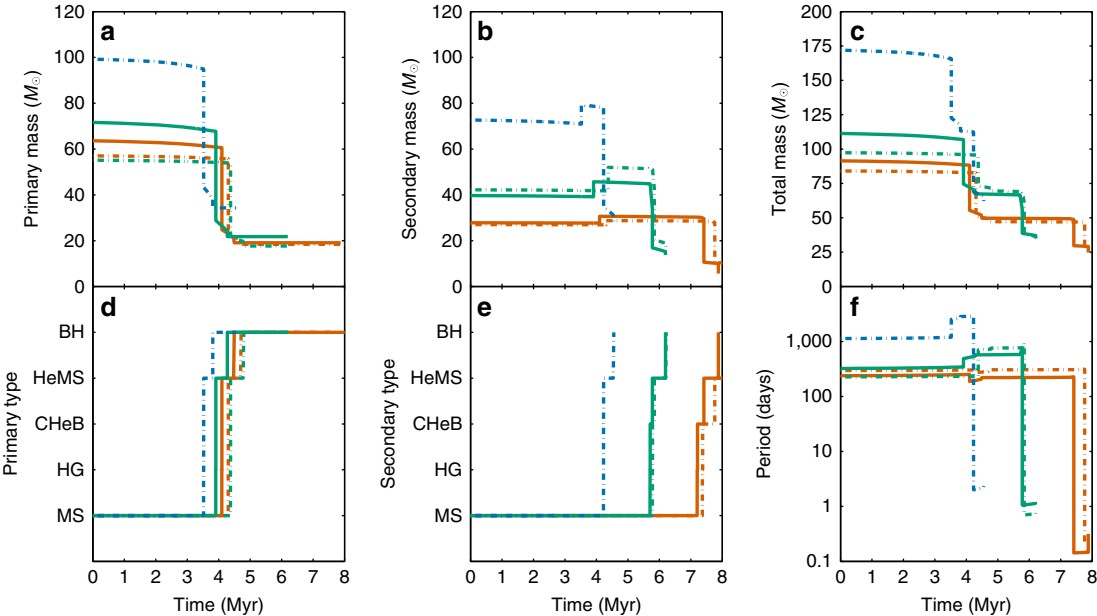

**Figure 2 | Typical evolution of BBHs progenitors.** Evolution in time of representative GW150914 (blue), GW151226 (orange) and LVT151012 (green) progenitors at 10% solar ($Z = 0.002$, solid lines) and 5% solar metallicity ($Z = 0.001$, dashed lines). (**a**) The mass of the initially more massive star. The stars lose mass through stellar winds, mass transfer and supernovae. (**b**) The mass of the secondary star. The stars may accrete mass during mass transfer episodes. (**c**) The evolution of the total mass of the binary. (**d**) The evolutionary stage (stellar type) of the initially more massive star as given by Hurley *et al.*[39] (see Results for definitions). (**e**) The evolutionary stage (stellar type) of the secondary star. (**f**) The orbital period of the binary in days.

Introduction to be comparable. Other proposed formation scenarios struggle to produce at least one of the observed BBHs. For example, both chemically homogeneous evolution[11–13] and dynamical formation in old, low-metallicity globular clusters in the model of Rodriguez *et al.*[14] (see their Fig. 2) have little or no support for relatively low-mass BBHs such as GW151226, which has a total mass $M = 21.8 \pm^{5.9}_{1.7} M_\odot$ (ref. 5). The ability of a single channel to explain all observed events will be tested with future GW observations[5,30].

We form $\sim 2 \times 10^4$ BBHs that merge in a Hubble time per $1 \times 10^9$ solar masses of star formation at 10% solar metallicity in our Fiducial model, using the Kroupa[31] initial mass function (IMF), a uniform mass ratio distribution and assuming that all stars are in binaries. This increases to $\sim 3 \times 10^4$ BBHs per $1 \times 10^9$ solar masses of star formation at $Z = 5\% Z_\odot$. Rescaling by the total star formation rate[32] at redshift $z = 0$, this would correspond to a BBHs formation rate of $\sim 300 \, \text{Gpc}^{-3}$ per year, assuming all star formation happens at 10% solar metallicity. This can be compared to the empirical LIGO BBHs merger rate estimate[5] of $9–240 \, \text{Gpc}^{-3}$ per year. However, this comparison should be made with caution, because even local mergers can arise from binaries formed at a broad range of redshifts and metallicities. An accurate calculation of the merger rate requires the convolution of the metallicity-specific redshift-dependent star formation rate with the time delay distribution, integrated over a range of metallicities[33].

There are many uncertainties in the assumptions we make (see Methods for details of our default assumptions). The evolution of massive progenitor binaries is poorly constrained by observations, although there has been recent progress, such as with the VLT-FLAMES Tarantula Survey in the 30 Doradus region of the Large Magellanic Cloud[34].

In rapid population synthesis codes such as COMPAS, these uncertainties are treated by parametrizing complex physical processes into simple one or two parameter models, such as treating the CE with the $\alpha$ prescription[35], or scaling Luminous Blue Variable (LBV) mass loss rates with $f_{LBV}$. The

| Time (Myr) | $M_1$ ($M_\odot$) | $ST_1$ – | | $ST_2$ – | $M_2$ ($M_\odot$) | $a$ ($R_\odot$) |
|---|---|---|---|---|---|---|
| 0.0 | 63.6 | MS | 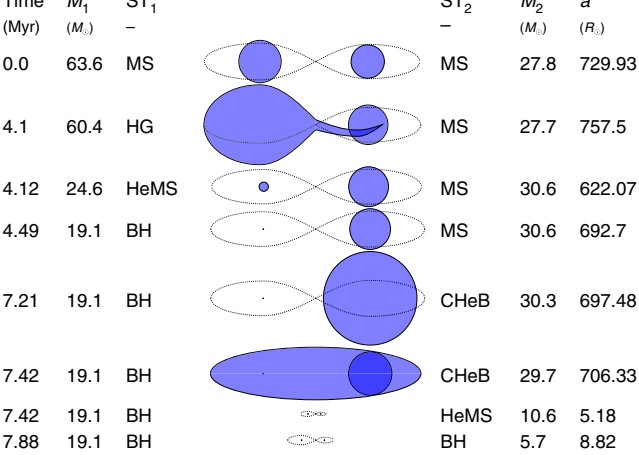 | MS | 27.8 | 729.93 |
| 4.1 | 60.4 | HG | | MS | 27.7 | 757.5 |
| 4.12 | 24.6 | HeMS | | MS | 30.6 | 622.07 |
| 4.49 | 19.1 | BH | | MS | 30.6 | 692.7 |
| 7.21 | 19.1 | BH | | CHeB | 30.3 | 697.48 |
| 7.42 | 19.1 | BH | | CHeB | 29.7 | 706.33 |
| 7.42 | 19.1 | BH | | HeMS | 10.6 | 5.18 |
| 7.88 | 19.1 | BH | | BH | 5.7 | 8.82 |

**Figure 3 | Formation of GW151226.** Typical formation of GW151226 at 10% solar metallicity in our model, as described in the Results. The columns show the time, the masses and stellar types of the primary and secondary, $M_1$, $ST_1$ and $M_2$, $ST_2$ respectively, and the semi-major axis $a$. Some intermediate stages of the evolution are omitted for clarity.

multidimensional space of model parameters, including $\alpha$ and $f_{LBV}$, must then be explored, to properly examine the model uncertainties.

We leave a full exploration of this parameter space for future studies with COMPAS; here we follow the common approach[27,36,37] of varying individual parameters independently and assessing their impact relative to the Fiducial model.

In the Fiducial model, we used the 'delayed' supernova model of Fryer *et al.*[38]. We have also checked that using the 'rapid' model of Fryer *et al.*[38] does not significantly alter the typical evolutionary pathways for forming heavy BBHs discussed here, as both models predict high-mass BH formation through almost complete fallback.

Mennekens and Vanbeveren[36] use a LBV mass loss rate of $10^{-3} M_{\odot}$ per year. They find that such strong mass loss can shut off the typical channel for BBHs formation. In COMPAS, we follow SSE[39] for identifying LBVs as massive stars with $L/L_{\odot} > 6 \times 10^5$ and $(R/R_{\odot})(L/L_{\odot})^{1/2} > 10^5$. We find that increasing the mass loss rate of LBVs from $1.5 \times 10^{-4} M_{\odot}$ to $10^{-3} M_{\odot}$ per year does not significantly change the total BBHs merger rate; nevertheless, the number of BBHs mergers similar to LVT151012 was reduced by a factor of $\sim 10$ for progenitors at 5% solar metallicity.

In the Fiducial model, we only permit evolved CHeB stars with a well defined core-envelope separation to survive CE events (see Methods). This model therefore corresponds to the pessimistic model of Dominik et al.[27], which is also the standard model (M1) of Belczynski et al.[7]. We also consider an alternate model where we allow HG donors to initiate and survive CE events, as in the optimistic model of Dominik et al.[27] We find that the optimistic CE treatment predicts total BBHs merger rates, which are $\sim 3$ times higher than the Fiducial model at $Z = 10\% Z_{\odot}$ and $\sim 2$ times higher at $Z = 5\% Z_{\odot}$. This optimistic variation also raises the total merging BBHs mass that can be formed at a given metallicity; for example, at $Z = 10\% Z_{\odot}$, the maximum total BBHs mass rises from $\sim 50 M_{\odot}$ for the pessimistic model to $\sim 60 M_{\odot}$ for the optimistic model, as also noted by Dominik et al.[27] The spread between these optimistic and pessimistic models also reflects the uncertainty in the radial evolution of very massive stars; the results of the pessimistic model could move toward those of the optimistic model if the radial expansion for the most massive stars predominantly happens during the CHeB phase rather than during the HG phase.

For a very small number of our simulated systems, immediately after the CE is ejected the binary comprising a BH and a HeMS secondary that is already overfilling its Roche lobe. In the Fiducial model we treat these systems as an unsuccessful CE event, leading to mergers. Similar studies[40,41] have allowed only those systems which overfill the Roche lobe by no more than 10% at the end of the CE phase to survive. We also consider the extreme alternative of allowing all such systems to survive. The HeMS stars lose a significant fraction of their mass through rapid but stable mass transfer onto the BH companion. Most of this mass is removed from the binary as the BH companion can only accrete at the Eddington limit and the HeMS star leaves behind a relatively low mass BH. We verify that this has no impact on our conclusions.

We test the impact of the assumed CE ejection efficiency by changing the value of $\alpha \lambda$ from the fiducial 0.1 to 0.01. At 10% solar metallicity we find the total BBHs merger rate drops by a factor of $\sim 2$. Dominik et al.[27] performed the same study, setting $\alpha \lambda = 0.1$ (model V2) and $\alpha \lambda = 0.01$ (model V1), and report the same decrease (see Tables 1, 2 and 3 in Dominik et al.[27]). At 5% solar metallicity, the total BBHs merger rate drops by a factor of $\sim 4$, with the specific merger rates of binaries like GW151226, LVT151012 and GW150914 dropping by factor of $\sim 25$, $\sim 4$ and $\sim 50$, respectively. The maximum BBHs mass produced at 10% solar metallicity increases from $\sim 50 M_{\odot}$ in the Fiducial model to $\sim 60 M_{\odot}$ under this variation. At 5% solar metallicity we find that the maximum total BBHs mass decreases from $\sim 75 M_{\odot}$ to $\sim 65 M_{\odot}$.

In conclusion, we have shown that GW150914, GW151226 and LVT151012 are all consistent with formation through the same classical isolated binary evolution channel via mass transfer and a common envelope. GW observations can place constraints on the uncertain astrophysics of binary evolution[42–46]. Although the focus of this paper has been on the constraints placed by the observed BBHs masses, other observational signatures, including merger rates (and their variation with redshift)[47], BH spin magnitude and spin-orbit misalignment measurements[48–50] and possibly a GW stochastic background observation[51,52], can all contribute additional information. COMPAS will provide a platform for exploring the full evolutionary model parameter space with future GW and electromagnetic observations.

## Methods

**COMPAS population synthesis code.** COMPAS includes a rapid Monte-Carlo binary population synthesis code to simulate the evolution of massive stellar binaries, the possible progenitors of merging compact binaries containing neutron stars (NSs) and BHs which are potential GW sources. Our approach to population synthesis is broadly similar to BSE[53] and the codes derived from it, such as binary_c[54–57] and StarTrack[58,59].

COMPAS was developed to explore the many poorly constrained stages of binary evolution, such as mass transfer, CE evolution and natal supernova kicks imparted to NSs and BHs[25]. Here we provide a brief overview of our default assumptions.

For our Fiducial model, we simulate probable BBHs progenitor binaries with the primary mass $m_1$ drawn from the Kroupa IMF[31] up to $m_1 \leq 100 M_{\odot}$ where the IMF has a power-law index of $-2.3$. The mass of the secondary is then determined by the initial mass ratio $q \equiv m_2/m_1$, which we draw from a flat distribution between 0 and 1 (ref. 60).

The semimajor axis $a$ is chosen from a flat-in-the-log distribution[61,62] and restricted between $0.1 < a/\text{AU} < 1,000$; the period distribution is therefore set by the convolved semimajor axis and mass distributions. The boundaries on the component masses and separations are chosen to safely encompass all individual solutions yielding BBHs of interest and so impact normalization only. Binaries are assumed to have an initial eccentricity of zero; the initial semimajor axis distribution serves as a proxy for the periapsis distribution, which is the relevant parameter affecting binary evolution[37]. Stellar rotation and tides are not included in the Fiducial model.

We use the analytical fits of Hurley et al.[39] to the models of Pols et al.[28] for single stellar evolution. We note that the original grid of single star models extends only to 50 solar masses. We extrapolate above this limit, as described in Hurley et al.[39]

We include mass loss due to stellar winds for hot O stars following the Vink model[22,63], with a LBV mass loss rate of $f_{\text{LBV}} \times 10^{-4} M_{\odot}$ per year, independent of metallicity. In the Fiducial model $f_{\text{LBV}} = 1.5$ (ref. 22). For Wolf-Rayet stars, we use the formalism of Hamann and Koesterke[64], modified as in Belczynski et al.[22] to be metallicity dependent ($\propto Z^{0.85}$) based on Vink and de Koter[65]. We assume that all stellar winds are isotropic and remove the specific angular momentum of the mass losing object. We do not account for wind accretion by a companion.

Mass transfer occurs when the donor star fills its Roche lobe, whose radius is calculated according to Eggleton[66]. Although all of our binaries are initially circular, supernovae can lead to some eccentric systems. We use the periastron to check whether a star would fill its Roche lobe, whose radius is computed for a circular orbit with the periastron separation. We assume that mass transfer circularizes the orbit.

In the absence of accurate stellar models spanning the full parameter space of interest, we use a simplified treatment of mass transfer. We assume that mass transfer from main-sequence, core-hydrogen-burning donors (case A) is dynamically stable for mass ratios $q \geq 0.65$. We follow deMink[57] and Claeys et al.[67] in assuming that case A systems with $q < 0.65$ will result in mergers as the accretor expands and brings the binary into contact[40]. Stable case A mass transfer is solved using an adaptive algorithm[68], which requires the radius of the donor to stay within its Roche lobe during the whole episode; when this is impossible, we assume that any donor mass outside the Roche lobe is transferred on a thermal timescale until the donor is again contained within its Roche lobe. In our Fiducial model we first test whether mass transfer is stable; if it is, we treat stable mass transfer from all evolved stars (case B or case C) equally, without distinguishing between donors with radiative and convective envelopes: we remove the entire envelope of the donor on its thermal timescale[69]. We follow Tout et al.[70] and Belczynski et al.[59] in our model for the rejuvenation of mass accreting stars.

The efficiency of mass transfer (that is, how conservative it is) is set by the rate at which the accretor can accept material from the donor. For NS and BH accretors, the maximum rate of accretion is defined by the Eddington limit. We assume that a star can accrete at a rate $C M_{\text{acc}}/\tau_{\text{th}}$ with the Kelvin–Helmholtz thermal timescale $\tau_{\text{th}} = G M M_{\text{env}}/RL$, where $G$ is the gravitational constant, $M$ is the total mass of the star, $M_{\text{env}}$ is the mass of the envelope, $R$ is the radius of the star and $L$ is its luminosity. The constant $C$ is a free parameter in our model; we use $C = 10$ for all accretion episodes in the Fiducial model[53]. The material that fails to be accreted is removed from the system with the specific angular momentum of the accretor via isotropic re-emission.

We determine the onset of dynamically unstable mass transfer by comparing the response of the radius of the donor star to a small amount of mass loss against the response of the orbit to a small amount of mass transfer[71]. We use fits to condensed polytrope models[71,72] to calculate the radius response of a giant to mass loss on a dynamical timescale. Dynamically unstable mass transfer leads to a CE. If the donor star is on the HG, we follow Belczynski et al.[7,73] in assuming such

systems cannot survive a CE. In fact, such systems may never enter CE at all. Pavlovskii et al.[74] have shown that in many cases mass transfer from HG donors will be stable and not lead to a CE.

All of our successful CE events therefore involve a donor star which has reached CHeB. For CE events, the $\lambda$ parameter, which characterizes the binding energy of the envelope[35], is set to $\lambda = 0.1$ (refs 7,27,75,76), whereas the $\alpha$ parameter, which characterizes the efficiency of converting orbital energy into CE ejection, is set to $\alpha = 1$. If one of the stars in the post-CE binary is filling its Roche lobe immediately after CE ejection, we assume that there is insufficient orbital energy available to eject the envelope and the binary evolution is terminated in a merger. We assume that CE events with successful envelope ejections circularise orbits (see section 10.3.1 of Ivanova et al.[26]).

The relationship between the pre-supernova core mass and the compact remnant mass follows the 'delayed' model of Fryer et al.[38]. Supernova kicks are assumed to be isotropic and their magnitude is drawn from a Maxwellian distribution with a one-dimensional velocity dispersion $\sigma = 250\,\mathrm{km\,s^{-1}}$ (ref. 77), reduced by a factor of $(1 - f)$, where $f$ is the fallback fraction, calculated according to Fryer et al.[38]. As in Belczynski et al.[7], we find that most of our heavy BHs form through complete fallback without a supernova or associated kick.

**Data availability.** We make the results of our simulations available at http://www.sr.bham.ac.uk/compas/.

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

## Acknowledgements

We thank Chris Belczynski, Christopher Berry, Natasha Ivanova, Stephen Justham, Vicky Kalogera, Gijs Nelemans, Philipp Podsiadlowski, David Stops and Alberto Vecchio for useful discussions and suggestions. I.M. acknowledges support from STFC grant RRCM19068.GLGL; his work was performed in part at the Aspen Center for Physics, which is supported by National Science Foundation grant PHY-1066293. A.V.G. acknowledges support from CONACYT. S.S. and I.M. are grateful to NOVA for partially funding their visit to Amsterdam to collaborate with S.d.M. S.d.M. acknowledges support by a Marie Sklodowska-Curie Action (H2020 MSCA-IF-2014, project id 661502) and National Science Foundation under Grant Number NSF PHY11-25915.

## Author contributions

All authors contributed to the analysis and writing of the paper.

## Additional information

**Competing interests:** The authors declare no competing financial interests.

**Publisher's note**: 

