## [Peer Review File · Nature Communications]

Reviewer #1 (Remarks to the Author):

My first question on reading this paper is what particular new insight does it offer that would justify inclusion in Nature Communications? I am not sure that this question has been answered.

The manuscript presents an analysis of the theoretical and model pathways that could lead to gravitational wave transient events such as the three sources (two firm detections, one candidate) identified in the first LIGO science run. It confirms that all three events can be produced by the authors' binary synthesis code and extrapolates the properties of the progenitor black hole binaries.

This is interesting and worthy of publication in the astronomical literature - however I am not certain that it has the level of originality required for a journal in the Nature family. There have been several predictions and analyses based on other stellar population synthesis codes published in the last six months (e.g. Belczynski et al, Eldridge et al, Lipunov et al - all of which the authors reference and discuss) and the current manuscript largely agrees with the results of those papers. Given the significant uncertainties inherent in binary population synthesis, this independent analysis adds to the literature and provides a useful check of the interpretation of these events, but it is not clear to me that it significantly advances our understanding of the progenitor systems of these events, or of gravitational wave sources more generally. In fact, the paper reads largely as a verification test and promotion of the COMPAS code (which is itself a valid thing to publish, but again I'm unclear whether this satisfies the scope of a Nature-family journal).

The manuscript is, on the whole, well written and reflects a thorough and reliable analysis of the data and relevant models. The authors choices of fiducial models are well explained and justified. The figures are clear and well labelled (although they could have been of higher resolution) I have no quibble with the scientific merit of the work which appears to be high, and well presented.

My one scientific query would be to question the time resolution of the models - unless I overlooked this, it is not stated, and I would like the authors to comment on how reliable the abrupt transitions in mass and period (as demonstrated in figure 2) are. I realise that many of the processes concerned are rapid, but are the jumps in source properties really as abrupt as shown?

In conclusion, I believe this paper is of scientific merit and worthy of publication in the academic literature with minor or no revisions - however I believe that an editorial decision regarding the degree of originality and new insight required for a paper in this journal is necessary.

Reviewer #2 (Remarks to the Author):

Comments to the authors:

This is an interesting paper about formation channels of binary BHs such as the LIGO events. I have suggestions and comments about the manuscript.

Main comments:

(1)

The authors assume that mass transfer from all evolved stars is stable. But, this assumption seems crucial because a convective star is commonly believed to lead unstable mass transfer due to the M-R relation of the stellar envelope. Although Pavlovskii et al. (2016) has discussed the possibility that mass transfer might be more stable than expected before, the treatment of mass transfer is still highly uncertain. Therefore, I'd like to suggest the authors to run population

synthesis simulations with distinguishing properties of donor's envelope (radiative or convective) in the same way as in most of previous works, and then compare the results to the author's original results.

(2)

Mass transfer rate determines the results of this paper (e.g. transferred mass). Does the prescription the authors adopt produce mass transfer rates consistent with previous studies, e.g. Paczynski & Sienkiewicz (1972) or Ritter (1988)? I'd like to suggest the authors to discuss this.

(3)

Population synthesis models are useful to understand typical evolutionary pathway of binaries and give statistical arguments rather than individual binary evolution. Because there are a lot of free parameters in their models (e.g. mass loss, CE, etc.), it doesn't sound difficult to explain properties of only two merging binary BHs. I'd like to suggest the authors to discuss whether these formation channels the authors discuss are general outcomes from binaries with $Z=Z_{\text{sun}}$ and $5\%Z_{\text{sun}}$ or very rare cases among those to form binary BHs.

(4)

The authors conclude that their fiducial model can form binary BHs, which are consistent with the LIGO observations, in terms of the masses and separations. But, the LIGO gave some constraints on their spins as well.

Although the authors don't discuss spins of BHs (and stars), it would be useful to discuss this aspect. For example, Kushnir et al 2016 (MNRAS 462, 844) have discussed formation of binary BHs, consistent with the spin values.

(5)

This comment is relevant to (4). The authors don't consider tidal effects. I wonder if the assumption is good or not, because a metal-enriched star generally has a large radius and then the stellar spin is likely to increase by tidal forces. I'd like to suggest the authors to justify this assumption.

(6)

The models and results seem similar to those in previous works, e.g. Belczynski and the collaborators. I'd like to suggest the authors to give some discussions about comparison with the previous results and clarify new points in this paper more explicitly.

(7)

The authors have left an exploration of the parameter space for future studies. I understand that parameter studies for all uncertainties would be difficult. However, I'd like to suggest the authors to check the dependence of their results on the CE parameter (combination of $\alpha * \lambda$) because formation rate of binary BHs is very sensitive to these parameters. Dominik et al. (2012) have found that the rate differs by a few orders of magnitude.

Minor comments:

(8)

The authors assume that when the accretor is a star, the maximum rate is limited at $C * M_{\text{acc}} / t_{\text{th}}$ and $C=10$.

I'd like to suggest the authors to justify these two assumptions or cite some literatures.

(9)

I'd like to suggest the authors to overplot the contours of BH mass estimates shown in Fig.4 of Abbott et al. 2016 (<http://arxiv.org/abs/1606.04856>) to compare to the author's results in Fig.1.

Referee 1

My first question on reading this paper is what particular new insight does it offer that would justify inclusion in Nature Communications? I am not sure that this question has been answered.

The manuscript presents an analysis of the theoretical and model pathways that could lead to gravitational wave transient events such as the three sources (two firm detections, one candidate) identified in the first LIGO science run. It confirms that all three events can be produced by the authors' binary synthesis code and extrapolates the properties of the progenitor black hole binaries.

This is interesting and worthy of publication in the astronomical literature - however I am not certain that it has the level of originality required for a journal in the Nature family. There have been several predictions and analyses based on other stellar population synthesis codes published in the last six months (e.g. Belczynski et al, Eldridge et al, Lipunov et al - all of which the authors reference and discuss) and the current manuscript largely agrees with the results of those papers. Given the significant uncertainties inherent in binary population synthesis, this independent analysis adds to the literature and provides a useful check of the interpretation of these events, but it is not clear to me that it significantly advances our understanding of the progenitor systems of these events, or of gravitational wave sources more generally. In fact, the paper reads largely as a verification test and promotion of the COMPAS code (which is itself a valid thing to publish, but again I'm unclear whether this satisfies the scope of a Nature-family journal).

As the referee quite rightly points out, and as we have acknowledged in our paper, our manuscript follows some 40 years of theoretical work in the field of binary evolution predicting the production of merging compact-object binaries. Of course, following the recent direct detection of gravitational waves, the field has undergone a revolutionary transformation, and these detections provide an unprecedented opportunity of confronting theory with observations.

We, for the time, have shown that all three gravitational-wave events (GW150914, GW151226, and LVT151012) can be generically formed through a single physical channel, without the need for fine-tuning the assumptions. We demonstrated that this conclusion is robust against variations in the main uncertain process, namely, common-envelope physics.

Another new result is the difficulty of forming unequal mass merging binary black holes at solar metallicity; however, it is possible to form merging binary black holes with more extreme mass ratios at lower metallicities. This is important because the mass ratio has been considered a possible indicator of the formation channel. For example, the LIGO-Virgo collaboration paper describing the full O1 binary black hole search results (arXiv:1606.04856) states: "Isolated binary evolution is thought to prefer comparable masses, with mass ratios $q < 0.5$ unlikely for the classical scenario (Dominik et al., 2012) and implausible for chemically homogeneous evolution (de Mink and Mandel, 2016). The dynamical formation channel also prefers comparable masses, but allows for more extreme mass ratios; observations of merging binary black holes with extreme mass ratios could therefore point to their dynamical origin." On the contrary, our finding shows that significantly unequal masses can be generated through the classical isolated binary evolution channel provided the formation happens at low metallicity.

Our contribution is timely and we believe it has wider relevance to the community.

The manuscript is, on the whole, well written and reflects a thorough and reliable analysis of the data and relevant models. The authors choices of fiducial models are well explained and justified. The figures are clear and well labelled (although they could have been of higher resolution) I have no quibble with the scientific merit of the work which appears to be high, and well presented.

We thank the referee for these kind comments.

My one scientific query would be to question the time resolution of the models - unless I overlooked this, it is not stated, and I would like the authors to comment on how reliable the abrupt transitions in mass and period (as demonstrated in figure 2) are. I realise that many of the processes concerned are rapid, but are the jumps in source properties really as abrupt as shown?

We resolve the physics on the relevant timescale, down to the thermal timescale. The rapid transitions in mass and period are typically episodes of case B or case C mass transfer, which occur on a thermal timescale (a few hundred or thousand of years) and a dynamical timescale (a few hours or days), respectively. In the plots these appear as abrupt jumps. There are also evolutionary phases which we really do treat as occurring instantaneously, namely stellar type changes, the common envelope phase and supernovae. This is valid as these occur on timescales much shorter than the evolutionary timescales.

In conclusion, I believe this papers is of scientific merit and worthy of publication in the academic literature with minor or no revisions - however I believe that an editorial decision regarding the degree of originality and new insight required for a paper in this journal is necessary.

Referee 2

This is an interesting paper about formation channels of binary BHs such as the LIGO events. I have suggestions and comments about the manuscript.

Main comments:

(1) The authors assume that mass transfer from all evolved stars is stable. But, this assumption seems crucial because a convective star is commonly believed to lead unstable mass transfer due to the M - R relation of the stellar envelope. Although Pavlovskii et al. (2016) has discussed the possibility that mass transfer might be more stable than expected before, the treatment of mass transfer is still highly uncertain. Therefore, I'd like to suggest the authors to run population synthesis simulations with distinguishing properties of donor's envelope (radiative or convective) in the same way as in most of previous works, and then compare the results to the author's original results.

We allow for unstable mass transfer from evolved stars that leads to common envelope events. We apologize if this was not clear. The text now reads:

"In our Fiducial model we first test whether mass transfer is stable; if it is, we treat stable mass transfer from all evolved stars (case B or case C) equally, without distinguishing between donors with radiative and convective envelopes: we remove the entire envelope of the donor on its thermal timescale (Kippenhahn & Weigert 1967)."

The referee may also have been concerned about our assumption regarding the common envelope; namely whether "Hertzsprung gap" stars can initiate and survive a common envelope. We tested the effect of varying this assumption, and the result is discussed in the text in Section 4.

(2) Mass transfer rate determines the results of this paper (e.g. transferred mass). Does the prescription the authors adopt produce mass transfer rates consistent with previous studies, e.g. Paczynski & Sienkiewicz (1972) or Ritter (1988)? I'd like to suggest the authors to discuss this.

We have conducted extensive code comparisons between COMPAS and alternative population synthesis codes, including binaryc and StarTrack, the latter of which adopts the Ritter (1988) mass transfer prescription. We find that the mass donation rates are consistent within numerical uncertainties. That said, the fraction of the transfer mass that is actually accreted is one of the key uncertainties (e.g., de Mink et al., 2007).

(3) Population synthesis models are useful to understand typical evolutionary pathway of binaries and give statistical arguments rather than individual binary evolution. Because there are a lot of free parameters in their models (e.g. mass loss, CE, etc.), it doesn't sound difficult to explain properties of only two merging binary BHs. I'd like to suggest the authors to discuss whether these formation channels the authors discuss are general outcomes from binaries with $Z=Z_{\odot}$ and $5\%Z_{\odot}$ or very rare cases among those to form binary BHs.

Figure 1: Observable chirp mass distribution for three metallicities, solar (blue), 5% solar (green) and 1% solar (red, not discussed in Stevenson+ 2016), weighted by LIGO selection effects. The vertical black lines denote the 90% measurement of the chirp mass for GW151226 (solid), LVT151012 (dashed), and GW150914 (dotted), from left to right.

We followed the referee’s suggestion and added the following sentence to the text in section 3:

“We find that the chirp mass of LVT151012 lies near the peak of the mass distribution of BBH mergers formed at solar metallicity which are observable by aLIGO. We also find significant support at the chirp mass of GW151226. There remains significant support for both systems at 5% solar metallicity. GW150914 cannot be formed at solar metallicity in our model, and remains in the tail of the total mass distribution at 5% solar, which is the highest metallicity at which we form significant numbers of all three event types in the fiducial model. Events like GW150914 are much more common at 1% solar metallicity.”

Here, we include a plot showing the observable chirp mass distribution predicted by our model at the two metallicities we describe in the paper, plus a lower one, 1% solar. To account for selection effects, the distribution is weighted by $M_c^{2.5}$, where M_c is the chirp mass, since the horizon distance for a binary black hole scales as $M_c^{5/6}$.

(4) *The authors conclude that their fiducial model can form binary BHs, which are consistent with the LIGO observations, in terms of the masses and separations. But, the LIGO gave some constraints on their spins as well. Although the authors don’t discuss spins of BHs (and stars), it would be useful*

to discuss this aspect. For example, Kushnir et al 2016 (MNRAS 462, 844) have discussed formation of binary BHs, consistent with the spin values.

There is a great deal of uncertainty about the fraction of angular momentum of the progenitor objects retained by black holes during their collapse. For example, as Kushnir et al. point out, rapidly rotating progenitors can have dimensionless angular momentum in excess of 1. Although it is clear that, in order to collapse to a black hole, which has a maximum dimensionless angular momentum of 1, such stars must lose a fraction of their angular momentum, the exact amount of angular momentum lost depends on the fallback details and is a matter of debate. Similarly, spin tilts during supernova are possible; while there is no evidence for spin tilts during the formation of a black hole, Farr et al. (2011) have pointed out that the spin of the secondary pulsar in the double pulsar system J0737-3039 must have undergone a tilt. Thus, while spins are a very interesting and important constraint, they deserve a thorough discussion; we are, in fact, in the process of writing a separate paper devoted to a consideration of spin misalignment angles as evidence for formation channels (see also Rodriguez et al. (2016), which appeared while this manuscript was under review), but this requires a careful discussion which cannot be compressed into the present manuscript. We added references to both Kushnir et al. and Rodriguez et al.

(5) This comment is relevant to (4). The authors don't consider tidal effects. I wonder if the assumption is good or not, because a metal-enriched star generally has a large radius and then the stellar spin is likely to increase by tidal forces. I'd like to suggest the authors to justify this assumption.

We agree with the referee that tides can play a significant role in massive binary evolution in general. The key impact of tides is to reduce the orbital separation, which further enhances the merger rate (see, e.g., the discussion in Section 6 of Dominik et al., 2015, cf. Mennekens and Vanbeveren (2014)). Our merger rates are therefore conservative. Furthermore, we account for the uncertainty in the evolution of orbital separation through other model variations, such as enhancing the LBV mass loss rates; these implicitly encompass the uncertainty in the treatment of tides.

(6) The models and results seem similar to those in previous works, e.g. Belczynski and the collaborators. I'd like to suggest the authors to give some discussions about comparison with the previous results and clarify new points in this paper more explicitly.

We expanded the discussion of the comparison to previous results in Section 3.

*(7) The authors have left an exploration of the parameter space for future studies. I understand that parameter studies for all uncertainties would be difficult. However, I'd like to suggest the authors to check the dependence of their results on the CE parameter (combination of $\alpha * \lambda$) because formation rate of binary BHs is very sensitive to these parameters. Dominik et al. (2012) have found that the rate differs by a few orders of magnitude.*

We have run an additional set of models where we vary the combination of alpha and lambda. Specifically, we have varied α from our fiducial value of $\alpha = 1$ to $\alpha = 0.1$, at both solar and 5% solar metallicity. We have added a paragraph discussing this to the paper.

*(8) The authors assume that when the accretor is a star, the maximum rate is limited at $C * M_{acc}/t_{th}$ and $C=10$. I'd like to suggest the authors to justify these two assumptions or cite some literatures.*

We follow Hurley et al. (2002) in this assumption, and have cited the paper here explicitly.

(9) Id like to suggest the authors to overplot the contours of BH mass estimates shown in Fig.4 of Abbott et al. 2016 (<http://arxiv.org/abs/1606.04856>) to compare to the author's results in Fig.1.

We have added dashed lines to Figure 1 (we have also reformatted Figure 1 slightly) showing the constraints we apply to consider an event similar to one of the GW events. These lines are the same as described in the text. We are unable to overplot the contours of BH mass estimates from Abbott et al 2016 as the posterior distributions are not publicly available.

Reviewer #1 (Remarks to the Author):

I have carefully reviewed the revised manuscript together with the response from the authors to my original comments, and to the comments of the second referee. I do not believe I have anything further to add to this discussion. The authors have carefully addressed points from both referees and their analysis appears robust and well presented. While I still somewhat question the novelty of the results in this work, the authors make a reasonable case for new insights to be learned from their models.

I would have no objection to seeing this paper published in Nature Communications in its current form.

Reviewer #2 (Remarks to the Author):

Thank you for following my suggestions to revise the manuscript. I have still one question about mass transfer rate.

>> (2)

> We have conducted extensive code comparisons between
> COMPAS and alternative population synthesis codes, including
> binary and StarTrack, the latter of which adopts the Ritter (1988)
> mass transfer prescription. We find that the mass donation rates
> are consistent within numerical uncertainties. That said, the
> fraction of the transfer mass that is actually accreted is one of
> the key uncertainties (e.g., de Mink et al., 2007).

The authors here treat mass transfer by removing the entire envelope of the donor star on its thermal timescale.

Then, this mass transfer rate is similar to those by alternative synthesis code, which adopt different prescriptions for mass transfer.

However, physical assumptions for each prescription seem different.

Could you explain why these give similar rates?

Referee 1

I have carefully reviewed the revised manuscript together with the response from the authors to my original comments, and to the comments of the second referee. I do not believe I have anything further to add to this discussion. The authors have carefully addressed points from both referees and their analysis appears robust and well presented. While I still somewhat question the novelty of the results in this work, the authors make a reasonable case for new insights to be learned from their models.

I would have no objection to seeing this paper published in Nature Communications in its current form.

We are grateful to the referee for this assessment.

Referee 2

Thank you for following my suggestions to revise the manuscript. I have still one question about mass transfer rate.

“We have conducted extensive code comparisons between COMPAS and alternative population synthesis codes, including binaryC and StarTrack, the latter of which adopts the Ritter (1988) mass transfer prescription. We find that the mass donation rates are consistent within numerical uncertainties. That said, the fraction of the transfer mass that is actually accreted is one of the key uncertainties (e.g., de Mink et al., 2007).”

The authors here treat mass transfer by removing the entire envelope of the donor star on its thermal timescale. Then, this mass transfer rate is similar to those by alternative synthesis code, which adopt different prescriptions for mass transfer. However, physical assumptions for each prescription seem different. Could you explain why these give similar rates?

A mass transfer treatment consists of two parts: the assumed rate of mass loss from the donor, and the assumed rate of mass accretion (i.e., the fraction of donated mass that is accepted by the other star). We checked that the rates of mass donation are consistent between codes. The mass accretion rates, which are highly uncertain as discussed in the paper, are intentionally different. For example, while StarTrack assumes that 50% of donated material is accreted, we use a more physically motivated model in which the accretion rate depends on the thermal timescale of the accretor.